# P4DM: Measure the Link Delay with P4

**DOI:** 10.3390/s22124411

**Published:** 2022-06-10

**Authors:** Amir Al Sadi, Davide Berardi, Franco Callegati, Andrea Melis, Marco Prandini

**Affiliations:** Department of Computer Science and Engineering, University of Bologna, 40136 Bologna, Italy; davide.berardi@unibo.it (D.B.); franco.callegati@unibo.it (F.C.); a.melis@unibo.it (A.M.); marco.prandini@unibo.it (M.P.)

**Keywords:** SDN, P4, Link Delay Measurement

## Abstract

Network management strategies depend on a timely and accurate knowledge of the network performance measures. Among these, one of the most relevant is the delay of the links, which unfortunately is not easy to measure with accuracy, especially when considering multi-hop paths. This is a classical networking problem, for which several solutions have been proposed. Nonetheless, we argue in this manuscript that there is still some room for improving accuracy and effectiveness in the measurement. This paper proposes a new solution based on the exploitation of the P4 data plane programming language. The basic idea is to handle lightweight probe packets that are forged ad-hoc at the edge of a link and processed at the other edge. Hosts generate the probe packets that are then exploited by the P4 programs in the switches to implement the measure. This approach provides an accurate and reliable measure of the link transit time, also effective in multi-hop links. In this latter case, we show that the measurement is not influenced much by the packet loss when the network is overloaded, thus providing more reliable results with respect to more conventional tools such as the classical ping utility. The manuscript explains the proposed P4 solution; then, it provides a comparison with several other approaches found in the literature, showing that outperform most of them, and finally show the behavior of the proposed methodology when facing a multi hop network path on a congested network to prove its robustness.

## 1. Introduction

Measuring the delay of a link is challenging and its definition depends on network configuration, hindering the development of a uniform solution that may be applied to heterogeneous network layouts.

Many network management scenarios rely on an efficient and precise measurement of the delay on a link, called Link Delay Measurement (LDM) in the following. Some examples are managing quality-of-service, detecting security attacks, and supporting many other time-sensitive applications, as described, among others, by Cordeiro et al. [1] which uses LDM to provide quality-of-service for Mesh Networks, or by Al Sadi et al. [2], in which LDMs are used together with asymmetric flow measurement in order to detect a possible Denial-of-Service attack, or by Giridhar et al. [3], in which LDM is used to perform search efficiently in a clustered network. In newer terms, 5G aims at providing network slices tailored to specific vertical applications, among which the Ultra Reliable Low Latency for time sensitive services [4]. At the same time, the most advanced network architectures assumes the possibility to deploy specific service functions as close as possible to the final users in terms of network delay (the so called “edge computing” paradigm exemplified also in the Multi-access Edge Computing standard promoted by ETSI [5]). In all these cases a detailed knowledge of the delay on the links is part of what required to design the correct service solutions.

Understandably, then, the scientific community produced a considerable effort on LDM. Indeed the most precise and reliable the LDM the more effective the implementation of these functions. Unfortunately, achieving high-quality LDM is not trivial. In particular [2] illustrates that relying on a network level protocol utility is not an effective approach since links run between switches, that do not necessarily implement or interact with the IP protocol. Therefore, a general approach requires an algorithm that is able to directly work at the data plane level, in the layer 2.

One Way Delay [6] (from now on: OWD) is the LDM between two switches calculated considering one traffic direction only. An effective algorithm for the calculation of OWD can also be used for LDM by combining the results obtained for the forward and backward paths using the OWD measurement [7]. In this paper, we will focus on OWD delay calculation, which has been implemented in the past with techniques that can be divided into two main categories:Active: algorithms that rely on sending timestamped probes from sender to receiver, avoiding a possible bias by generating probes in a random way.Passive: algorithms that rely on measurement of traffic properties of the network. In this case the delay is calculated between a sender and a receiver that are not necessarily the source and destination of a traffic flow.

This work proposes *P4DM (P4 link Delay Measurement)*, an original probe-based approach for OWD measurement, based on state-of-the-art Software Defined Network [8]. *P4DM* exploits Data Plane Programmability implemented with the P4 switch programming languages. For the reader not fully familiar with P4, Kfoury et al. [9] recently published a survey on this topic providing a classification and taxonomy of a large number of articles, while also identifying future challenges and future perspectives. We will explain in the remainder of this manuscript which are the advantages of this approach and compare the performance of *P4DM* with other algorithms that already appeared in the literature.

*P4DM* is inspired by In-Network Telemetry (INT) [10], a telemetry framework designed to perform network-wide monitoring. By providing a large number of different telemetry parameters, INT is able to conduct monitoring tasks by enriching the packet with bytes of information. Following this paradigm, *P4DM* uses the level 3 IP options to store the information needed to perform the calculation.

The remainder of this paper is organized as follows. In Section 2, we review the relevant literature to present the background concepts we based our work upon, and to highlight the limitations of current solutions. In Section 3, we introduce the P4 programming paradigm and its integration with SDN architectures. In Section 4, we describe our P4-based solution to OWD measurement, defining the necessary data structures and the algorithms to manipulate them on the switches; Section 5 documents the results of testing their implementation. We draw conclusions in Section 6.

## 2. State of the Art

The scientific community has been interested to the OWD problem for a while. Nonetheless, the distributed control plane approach of the Internet does not provide good tools to this. The SDN paradigm, thanks to the introduction of the concept of a centralized control plane, fueled a new wave of interest into the topic, with particular reference to active strategies.

The literature that treats OWD calculation exploiting SDN is very scattered and diversified, but OpenFlow [11] is by far the most used protocol [12]. A complete overview of the most valuable solutions elaborated by the scientific community is given in [13], followed by a performance comparison of them.

A known OpenFlow-based approach to delay calculation [14,15] is to use a PacketOut message to let a switch inject a probe in the data plane, to measure the time needed for the reply to come back as a PacketIn. However, the time difference between sending the PacketOut and receiving the PacketIn is affected by many sources of unpredictable delays, such as the overhead of forwarding processing, and the OpenFlow messages transmission. For this reason, the OWD between the switches can be extracted only with a limited precision [15].

Generally speaking, OpenFlow introduces overhead when it parses messages to the data plane level, and this puts a strain on time-sensitive applications and compromises their efficiency.

Phoemius and Bouet propose Controller in the loop [16], an algorithm that exploits OpenFlow in an SDN-based environment by injecting OpenFlow probe packets in the data plane which gathers the timestamps of the sending and receiving time. The main drawback of this solution is that the OpenFlow messages cause more overhead then simple packet probes.

Altukhov and Chemeritsky propose Many Data Loops(MDL) [17], a variation of the Controller in the loop algorithm that eliminates the bias of the control channel latency by looping the probe many times between the switches. The main drawback of this solution is the overhead introduced by looping the probe a very high number of times, in the order of the thousands loops for one single RTT measurement.

OpenNetMon [15] is another solution that monitors flow metrics such as throughput, delay and packet loss, in OpenFlow networks. Using probes, OpenNetMon polls switches in an adaptive rate to establish the metrics.

Sinha et al. proposed TTL based looping [18] an improvement of the MDL algorithm by exploiting the TTL IP field. The controller injects the probe with a fixed TTL and every switch decrements it until it is 0. When the TTL is 0 the probe is sent back to the controller. This method suffers the same problems of the MDL solution.

Liao et al. proposed TTL based LLDP looping [19], similar to the Controller in the LLDP looping solution, which loops the probe between two switches in a hard-coded fashion using a custom LLDP format to calculate the OWD. This solution suffers from the overhead introduced by the LLDP flooding.

SLAM [20] is a latency monitoring framework that sends packet probes to switches and estimates the latency distribution based on the arrival timestamps of the control messages at the controller.

The works mentioned above are examples of approaches to calculate the OWD that with different flavor share the same drawback, i.e., relying on OpenFlow the probe information must travel back and forth to the control plane (the SDN controller) which adds overhead to the network and limit the accuracy of the measures.

For this reason in this work, we tackled the problem of active OWD measurement by exploiting Data Plane Programmability, in particular using the functionalities given by the P4 language [21]. P4 has some native features (such as registers, counters, etc.) that suit perfectly for the task. P4 has already been used to solve network management problems, also related to cybersecurity. An interesting P4 proof-of-concept has been proposed in [22] where authors demonstrated the feasibility of a detection and mitigation Explicit Congestion Notification (ECN) protocol abuse without any TCP protocol modification.

In a previous work [23], we investigated the opportunity to integrate an SDN OpenFlow based control plane with P4, to enhance the network monitoring capabilities. Some of the ideas presented in [23] regarding the exploitation of P4 for network management are here expanded and tailored to the specifics of the OWD problem.

## 3. Background about P4 and Its Integration with SDN

The idea of programmable switches has been around for a long time; in the past it was hindered by the performance degradation of programmable switches, due to the fact that the vendor chips had to adapt to different specifications instead of focusing on a subset of features and making them perform at their best. P4 is a programming language which lets the end users describe how the switch should process the packets. P4 exploits the concept of data plane programmability. By data plane programmability we intend to describe the ability of adding functionalities to the network and expose the packet processing logic to the control level in order to enable a systematic, fast and complete reconfiguration. Another important feature to effectively grant data plane programmability is flexibility. Flexibility is the capacity of changing network’s topology, resources, functions or services at will [24]. Given these information, we claim that P4 language is the best choice to describe and deploy network rules [25], to gather information (such as timestamps) from the devices and to perform a LDM [26]. A P4 switch differs from a traditional one since [27]:Data plane functionalities are unknown to the P4 switch. It is programmed to follow rules written in the P4 program.Control plane communicates with the data plane in a traditional fashion but tables and other data plane objects do not have fixed roles. The P4 compiler generates control plane APIs used to communicate with the data plane.

P4 introduces different abstractions that allow it to be a target-independent programming language. In particular:*Header types*: describe packet header formats.*Parsers*: describe header sequences of received packets, how to identify sequences for error checking and which headers and fields must be extracted from packets.*Tables*: which associate actions with user-defined keys. P4 tables generalize traditional switches ones and are able to exploit them for a large number of useful purposes.*Match-action units*: which execute the following operations:(1)Create lookup keys from packets fields or from metadata.(2)Execute table lookup using the obtained key, choosing an action to execute.(3)Execute the chosen action.*Control flow*: express an imperative program to process packets on a target.*Extern objects*: architecture-specific constructs that can be manipulated from P4 programs thanks to APIs, vendor-specific and not P4 programmable.*User-defined metadata*: user-defined metadata associated to packets.*Intrinsic metadata*: metadata associated to every packet available from the architecture. For example, the packet input port.

P4 enhances data plane programmability by adding desirable features to the data plane:*Flexibility*: packet forwarding policies are defined using code instead of using traditional fixed policies. Furthermore, it is possible to change network resources and functions by P4 code.*Expressiveness*: P4 is able to express complex hardware-independent packet processing algorithms only using general-purpose operations and table look-ups. Programs are portable into different targets that implement the same architecture.*Resource assignment and management*: P4 describes storage resources in an abstract way, compilers associate user defined fields to hardware resources and manages low level details such as allocation.*Software engineering*: P4 provides type checking, information hiding and software reuse.*Decoupling hardware and software evolution*: producers can use abstract architectures to decouple the evolution of low-level details from high level processing.*Debugging*: producers offer architectural software models to help developing and debugging P4 programs.

P4’s flexibility and efficiency are characteristics desired to design an high performance LDM calculation.

## 4. The Proposed Methodology and Solution

As already outlined the main contribution of this work is an algorithm that calculates the OWD of a network link, *P4DM*, implemented by means of suitable P4 programs installed in the network switches.

Two different scenarios will be considered in the following.

*Point to point.* There is only a single link between the SOURCE and the SINK node. In this scenario we can suppose the values of the OWD calculation are rather stable, since the propagation delay should be the dominant contribution to the absolute value.*Multi-hop.* The SOURCE and the SINK are linked by a number of nodes not known a-priori. In this configuration, we expect the values of the OWD calculation quite variable and *P4DM* behaves in a similar way to ping [28]. The advantage of the proposed solution is that it avoids the delay introduced by the processing in the end hosts, since it exploits P4 directly on the data plane.

### 4.1. Implementation

The goal of the implementation is to perform an OWD calculation by sending a dataplane probe between two connected switches. The measurement is performed between a node named SOURCE, the issuer of the OWD calculation and a SINK node which is the receiver of the probe. *P4DM* is a lightweight solution to calculate the LDM between two nodes without the need of synchronization. Not needing switches to be synchronized allows this solution to be used in any network that has two switches that speak P4.

Figure 1 describes the workflow of *P4DM*.

The state-of-the-art OWD measurements techniques are detailed in Section 5.2. They use a controller to send probes to the data plane. In our solution, there is no need for an SDN controller since P4 can fully function without it. However, for the sake of homogeneity, we included the SDN Controller in Figure 1. In this proposed measurement workflow, the probes are generated by an application (Controller like in Figure 1, or a host), which simulates the role of the controller in the other solutions. The switches add a timestamp to the probing packet every time it enters a queue.

### 4.2. Algorithm Design

The list of collected timestamps is summarized in Table 1.

The steps of *P4DM* are outlined in Figure 1. The algorithm proceeds as follows:1.The controller sends the packet probe to Switch 1.2.Switch 1 adds timestamps $T_{s1i\_m1}$ and $T_{s1e\_m1}$ to the packet while forwarding it towards Switch 2.3.Switch 2 adds timestamps $T_{s2i\_m2}$ and $T_{s2e\_m2}$ and forwards the packet to Switch 1.4.Switch 1 adds the packet timestamps $T_{s1i\_m3}$ and $T_{s1e\_m3}$ and forwards the packet to the Controller, which calculates the OWD between the Switch 1 and Switch 2 using the following equation:
$$\mathrm{OWD}=\frac{\left(T_{s1e\_m3}-T_{s1i\_m1}\right)}{2}-\frac{T_{diff\_s1\_m1}}{2}-\frac{T_{diff\_s2\_m2}}{2}-\frac{T_{diff\_s1\_m3}}{2}$$

The design of *P4DM* leverages the features of P4, namely three native capabilities that are not available in standard SDN switches:*Custom packet headers*: P4 is able to describe a completely customized packet header.*Timestamping*: P4 offers the possibility of retrieving the time of packet entry in switches’ ingress and egress queues at data level.*Custom forwarding rules and packet inspection*: P4 is able to inspect header fields, and to program forwarding rules based on the contents of custom headers.

Figure 2 illustrates the custom header we defined to implement *P4DM*.

The header is represented as a 24 bytes-wide frame. It is the concatenation of the Ethernet header (Source address, Destination address, EtherType) and:*Protocol ID - proto_id*: The ID of the protocol (e.g., the ID for IP is 0 × 0800) has been kept as it is in a standard Ethernet packet, even if it is not used by *P4DM*, to leave the frame alignment intact.*Destination ID - dst_id*: The port the packet will cross in the next hop.*Number of hop - nhop*: Number of hop crossed from the sender.*Timestamps*: Packet timestamps used to calculate the OWD, as listed in Table 1.

### 4.3. P4 Data Plane Features Exploited by *P4DM*

As previously mentioned, P4 is able to inspect and modify packet fields. The probe packets are identified by an EtherType set to a custom *myTunnel* value. When a probe packet is detected, the following actions are applied.

*Hop counting*. Every time a myTunnel packet crosses an ingress queue, the hop number *nhop* is incremented:

apply {
if (hdr.ipv4.isValid()) {
// Process only non-tunneled IPv4 packets
ipv4_lpm.apply();
if (hdr.myTunnel.isValid()) {
hdr.myTunnel.nhop = hdr.myTunnel.nhop + 1;
myTunnel_exact.apply();
}
}
}

*Forwarding*. The forwarding table applied to probe packets uses the *dst_id* field as a selection key. It forwards the packet through the port giving access to the link under analysis, and updates the *dst_id* field with the port which will be used on the receiving switch to send the packet back. The latter port identifier is obtained from a flow rule installed by the controller in advance of sending probes.

action myTunnel_forward(egressSpec_t port) {
standard_metadata.egress_spec = port;
hdr.myTunnel.dst_id = (bit<16>)port;
}

*Timestamping*. Timestamps are inserted in the packet when it is added to an egress queue, based on the hop counter. Since the first message from the controller is generated with *nhop*=1, the first encountered switch actually sees the packet as performing the second hop in the network. Consequently:(nhop=2) - $T_{diff\_s1\_m1}$ is set.(nhop=3) - $T_{diff\_s2\_m2}$ is set.(nhop=4) - $T_{diff\_s1\_m3}$ is set.

if (hdr.myTunnel.isValid()) {
if (hdr.myTunnel.nhop >= 2) {
if (hdr.myTunnel.nhop == 2) {
hdr.myTunnel.Tdiff_s1_m1 =
standard_metadata.egress_global_timestamp -
standard_metadata.ingress_global_timestamp;
}
if (hdr.myTunnel.nhop == 3) {
hdr.myTunnel.Tdiff_s2_m2 =
standard_metadata.egress_global_timestamp -
standard_metadata.ingress_global_timestamp;
}
if (hdr.myTunnel.nhop == 4) {
hdr.myTunnel.Tdiff_s1_m3 =
standard_metadata.egress_global_timestamp -
standard_metadata.ingress_global_timestamp;
}
}
}

At the application level, an SDN controller was implemented using ONOS. ONOS is one of the few network operating systems that integrates P4 with the application level, by implementing the P4 control plane level logic with the module developed by the P4Brigade project [29].

Figure 3 shows the three-tier architecture, in which P4 is used to program the behavior of components in the control and data planes, and by means of the native ONOS controller features at the application plane. The ONOS controller allows to:Extract semantics of information coming from the data plane.Show the calculated OWD in a dedicated web GUI.

This architecture showcases an application of the combined use of P4 and ONOS for network monitoring. This architecture was solely developed to simulate the use of *P4DM* in an environment similar to the other solutions and ONOS was not used in the tests. We argue that P4 can give a groundbreaking new perspective on network management and security, by enhancing data level information retrieval and description. Furthermore, assuming the availability of P4-enabled devices grants portability of solutions devised according to this architectural model to many physical or virtual targets, giving them a broad deployment scope.

**Figure 3 sensors-22-04411-f003:**
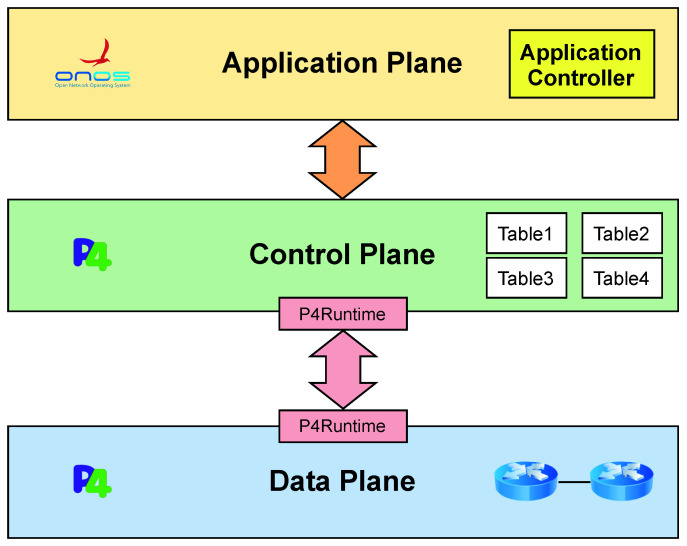
Three-tier SDN architecture.

## 5. Experimental Results

Three groups of tests were implemented, as outlined in Figure 4, to:compare *P4DM* with other solutions suggested in the literature (Section 5.2 and Section 5.3);compare *P4DM* with ping in different traffic scenarios (Section 5.4);assess the impact of the proposed solution on the packet time transmission of SDN switches (Section 5.5).

**Figure 4 sensors-22-04411-f004:**
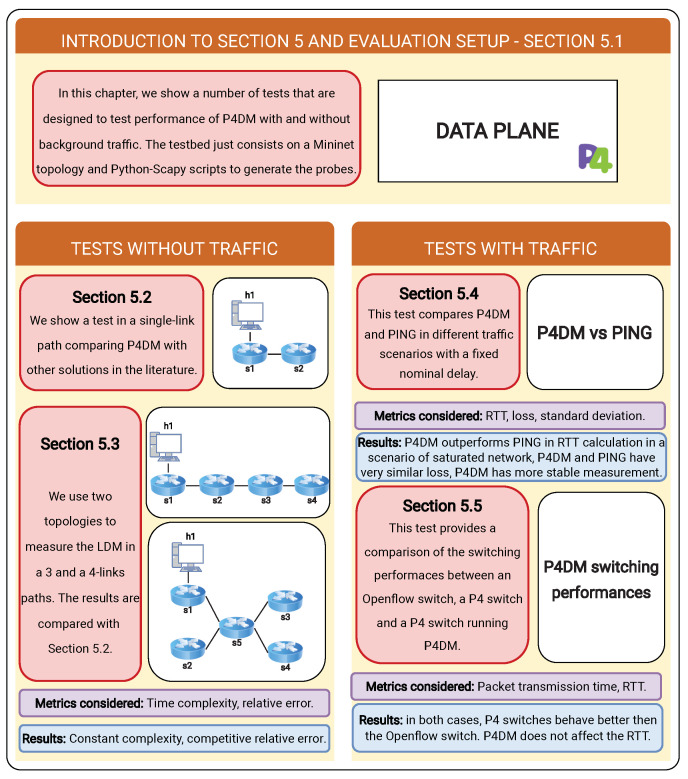
Graphical abstract of Section 5.

The tests conducted showed the potentials and performances of *P4DM*.

### 5.1. Experimental Setup

The following tests were performed using Mininet [30] in a Ubuntu 20.04 with 8 GB of RAM and an Intel i5 8th generation processor host. Figure 5 describes the infrastructure.

To simulate the traffic, Python and the Scapy network library were used to forge the probe packets with related custom IPv4 packets and to generate and receive probe packets. The collected probes timestamps were stored in the filesystem and the relevant statistics are then calculated by using a Python script. To improve proper confidence to the measures, batches of 50 probes are sent every 0.5s to avoid network congestion. We empirically noticed that with 50 packet probes a reliable calculation of the OWD can be obtained. Hence, we decided 50 probes as the fair amount to perform a *P4DM* measurement. In the following we will address with AV the average LDM obtained with *P4DM*. Finally, the measurements of the first two set of tests are performed in ideal conditions, with no background traffic and therefore with the network not congested apart from the test which compares *P4DM* with ping.

In order to recreate a test-bed similar to the algorithms considered in the comparison of the Section 5.2, the ONOS SDN controller [31] is included in the infrastructure. The infrastracture is based on two Docker [32] containers, the former hosting the Mininet topology and the latter hosting the ONOS operating system. However, for the tests related to *P4DM*, which does not need an SDN controller to work, we used the Mininet Linux command and the Scapy library to obtain a more lightweight implementation. The repository which contains the code used for the test section can be found in [33].

### 5.2. Tests for OWD Algorithm Comparison in a Link

*P4DM* was tested by setting a fixed delay in a network link and measuring it by sending a batch of packet probes. From now on, we will refer the above mentioned fixed delay as nominal delay (ND). Collected the batch, the average LDM (AV) is calculated. Since the path is a single link, we can assume that the delay is equal if calculated from SOURCE to SINK or vice versa. These tests were done using simple_switch [34], a P4 switch not meant for production which has the timestamp precision on the range of the microsecond. The tests evaluate the algorithm performances by considering two indicators.

*Relative Error (%)*: given ND, relative error is obtained as follows:
$$Error=\frac{AV-ND}{ND}\times 100$$*Time Complexity* This describes the time that it takes to run an algorithm. We have adopted the same methodology to estimate the time complexity as in [13].

The algorithms and the indicators used for this comparison are taken from works mentioned in the “State of the Art” section and from the review paper of D. Chefrour [13]. As already explained, given the nature of *P4DM*, the focus is on active measurement OWD solutions. These type of solutions, when based on SDN, use the SDN controllers to send packet probes on the network to retrieve the delay. The comparison described in this section only considers algorithms that use packet probes.

Table 2 reports the performance measures, with algorithms ordered by increasing time complexity. The solutions can be roughly grouped into 3 sets:*Constant complexity*: algorithms that have a O(1) complexity, like *Controller in the loop*, *Controller in the LLDP loop*, *Many data loops* for one path, *OpenNetMon*, *TTL based looping*, *TTL based LLDP looping* for one link; these algorithms report an error that spans in the range of 1–4% with a tested ND that is under 20 ms;*Linear complexity*: algorithms that have a O(n) complexity, like *Many data loops* for all links, *TTL based LLDP looping* for all links, *SLAM*; these algorithms report an error that spans in the range of 0.1–4% with a tested ND that is under 20 ms;*Superlinear complexity*: algorithms that have a O($n^{k}$) complexity, like *SdProber*; this algorithm does not report either the tested delay or the error.

**Table 2 sensors-22-04411-t002:** Algorithm comparison.

Technique	Nominal Delay	Error	Time Complexity
Controller in the loop	0–20 ms	1%	O(1)
Controller in the LLDP loop	5 ms	3%	O(1)
Many data loops	540 μs	3.7%	O(1) one path,
			O((n+e)(c+1)) all links
OpenNetMon	7 ms	2.3%	O(1)
TTL based looping	0–350 ms	n/r	O(1)
TTL based LLDP looping	1,	3,	O(1) one link,
	5,	0.4,	O(Kn) all links
	10 ms RTT	0.1%	
SLAM	0–20 ms	n/r	O(n)
SdProber	n/r	n/r	O($n^{3}$)
*P4DM*	10,	1.7,	O(1)
	50,	0.31,	
	100,	0.53,	
	200 ms	0.31%	

*P4DM* has a constant complexity since it only needs a packet to perform the measurement, putting it in the best time complexity group and the measurement accuracy is generally better than the compared solution. In particular, 4 fixed delays were tested:10 ms: which reports 1.7% compared to an average error of 3% for similar fixed delays. The only slightly better result is in *TTL based looping* which however has a worse complexity for more then a link.50 ms, 100 ms, 200 ms: which report errors that are less than 0.6%. This result is decidedly better given that the majority of solutions do not report errors for similar delays.

Given this information, we can now give a punctual comparison between the various algorithm and *P4DM*:*Controller in the loop*: this algorithm was tested in a 20 switches Mininet network and needs 2 packets per second sending probe rate to correctly calculate the OWD, to smooth the overhead introduced by the controller. Our algorithm achieves a very stable OWD calculation by only sending 5 successive probes. This gives the algorithm a notable advantage on the performance side. The authors of *Controller in the loop* do not show results on multi-links paths.*Controller in the LLDP loop*: this algorithm is very similar to *Controller in the Loop* but achieves a lightly worse performance. This algorithm only reports error on a single tested delay (5 ms) and for this reason it is not possible to compare the performances on higher OWDs. The algorithm was tested only on single link paths. Not many comparison other than the error value, that stands at 3% compared to the 1.7% of our solution, can be done between *P4DM* and this.*Many Data Loops*: this algorithm performs similarly to the other algorithms on a 540 μs RTT. This algorithm is also tested on a multi-link path. *P4DM*, as shown in the Section 5.3, maintains a constant complexity also in a multi-link scenario, while Many Data Loop has a linear time complexity.*OpenNetMon*: this algorithm performs similarly to the others. It was tested on a 4-switches Mininet network on multi-link paths. The probes rate is adaptive (F(throughput)) concerning the possibility of having a very expensive overhead to perform the OWD calculation, depending on the throughput of the network. *P4DM*, as shown in Table 2, maintains a reliable OWD calculation with 5 consecutive sent packets.*TTL based Looping*: Sinha et al. did not report many performance measurements of this solution.*TTL based LLDP looping*: this algorithm performs slightly better on the 5 ms and 10 ms RTT, achieving a sub 1% relative error. It was tested on a single-link path and reports a constant complexity, that degrades into a linear complexity in multi-link paths. The main drawback of this solution is the probes frequency rate, that spans from 1 to 100 packets per second. This characteristics can degrade the efficiency of the network in monitored paths. As previously underlined, our proposed algorithm achieves a stable OWD calculation by sending 5 consecutive packet probes.*SLAM*, *SDProber*: these 2 solutions do not report any accuracy estimation and have a worse time complexity then our solution, respectively linear and superlinear. Given the information reported, we claim that our algorithm performs better on every aspect.

The analysis generally reported a positive feedback on every considered performance aspect. Furthermore, it must be pointed out that *P4DM* needs a constant amount of 3 rules per switch, a lighter configuration cost compared to other solutions. Moreover it does not need a constant rate of probes per time unit, unlike most of the algorithm considered. This can be advocated to the ability of P4 to completely isolate the data plane and for this reason there is no need to tune the probes frequency to smooth the overhead introduced by interacting with the SDN controller.

### 5.3. P4DM in a Multiple Links Path

Tests were designed to verify the performances of *P4DM* in multi-links paths. Two network topologies were designed to test the algorithm on a 3-links path and a 4-links path.

Figure 6 shows the topology of the network in which a 3-links path was tested.

*P4DM* is applied to a linear path which starts from switch s1 and ends at switch s4. The controller is responsible to send and receive the packets.

Figure 7 shows the topology of the network in which a 3-links path was tested.

**Figure 6 sensors-22-04411-f006:**
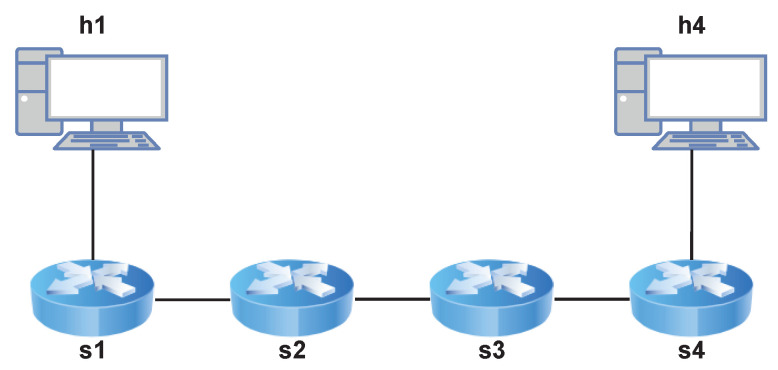
Network for the 3-links path OWD calculation.

This network allows to simulate a more challenging scenario for *P4DM*: a path starting from switch s1, going through switches s2, s5, s3 and ending at s4, was considered. Tests similar to those conducted to the one link case were performed. Table 3 shows a performance comparison between the 1-link case and these two additional cases.

Every link was set with the same nominal delay of the previous test and for this reason the nominal delay of the path is equals to:delay×number_of_links

The performance analysis registers a very small performance degradation. This can be advocated to 2 major factors:(1)The algorithm is designed to monitor the network link-by-link. To perform multiple-link paths OWD calculations both the packet header and the P4 application were modified, introducing a minimal overhead. Nonetheless, the performances are not sensibly worsened.(2)The virtualized environment introduces more noise if the path is composed by multiple links.

In conclusion, the tests underlined how *P4DM* can be effective on paths with more links as well as on a single link.

### 5.4. Comparison between Ping and P4DM in a Multi-Link Path Scenario

In this section, we compared *P4DM* and ping in a multi-link path scenario with variable traffic. This test is useful to evaluate the performances of our solution in the environment for which *P4DM* is designed for, comparing it with the most used utility to calculate the RTT between two hosts. We used the utility hping3 [35] to simulate different network congestion levels, by sending arbitrary 20 bytes-long headers with no data IP packets. Every RTT measurement was taken by sending 50 *P4DM* probes.

Figure 8 shows the comparison between ping and *P4DM* on a nominal delay of 60 ms RTT. In this case, we considered the RTT (as sum of the forward and backward delays) measured by *P4DM* instead of the OWD, to make it comparable with ping. This test compares the measured RTT by varying the packets per second injected in the network path. The topology analyzed is the one presented in Figure 6, using the same path considered in Section 5.3. However, to be able to perform the ping, a host was attached to s4. The chart shows how, apart from small fluctuations induced by the virtual environment, *P4DM* and ping perform similarly. However, after hitting the threshold of 1000 packets per second, *P4DM* outperforms ping by at least half the RTT. The difference between the two can be mainly advocated to the fact that *P4DM* is able to remove the queuing delay from the LDM measurements, as opposed to ping.

Table 4 shows the corresponding standard deviations on the measurements performed in Figure 8. The table shows that ping is more unstable in heavy traffic than *P4DM*, which outperforms ping since it is able to remove the queuing experienced in the switches from the RTT. In fact, the spike of both the standard deviation and RTT of ping is motivated by the high queuing time in the switches.

Packet loss has almost the same trend for both as shown in Figure 9. This is correlated with the ability of the network to forward IP packets: when saturated, more packets are dropped and concomitantly the loss increases.

This test demonstrates how *P4DM* behaves better in calculating the LDM then ping in heavy traffic, while keeping a better stability in measurements overall. The loss is roughly similar in all sorts of traffic, which is supported by the fact that *P4DM* is encapsulated in raw IP packets and ping in ICMP ones.

**Table 4 sensors-22-04411-t004:** *P4DM* and ping standard deviation.

Packets per Second	Standard Deviation of *P4DM*	Standard Deviation of Ping
100	0.14	0.63
200	0.02	0.1
500	0.01	0.07
800	0.02	0.08
1000	0.02	0.18
2000	0.4	21.56
5000	0.29	5.84
8000	0.06	9.90
10,000	0.18	23.4

**Figure 9 sensors-22-04411-f009:**
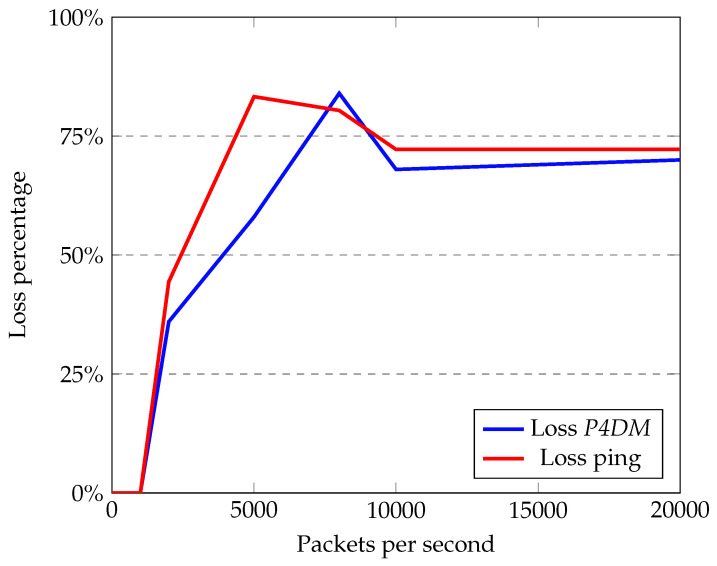
*P4DM* and ping loss comparison.

### 5.5. Impact of P4DM on the Switching Performance

This set of tests was designed to compare the packet forwarding performance.

In particular, three different running switches set-ups were considered:packet forwarding is OpenFlow based, using ONOS as an SDN controller;packet forwarding is P4 based, with no additional features;packet forwarding is P4 based, adding *P4DM* to the normal operations.

OpenFlow-enabled switches are the most used in SDN environments and P4-enabled switches require more computational resources to operate, giving a performance disadvantage to the second one on paper.

Figure 10 shows a packet transmission time comparison between switches using the different set ups. A calculation of Total Transmission Time of bursts of packets of length ranging between 5 and 200 was performed, for packet sizes of 512 Bytes and 8192 Bytes. The comparison clearly shows that providing P4 basic functionalities does not significantly degrades switches’ performances even in the scenario of switches running both P4 and *P4DM*. Overall, P4 does not introduce a sensitive performance overhead compared to OpenFlow.

Figure 11 shows a calculation of the average Round Trip Time (RTT) to transmit a set of 512 Bytes and 8192 Bytes ICMP packets. The goal of these tests was to examine the forwarding behaviour of the different switches (OpenFlow-enabled, P4-enabled and *P4DM*-enabled). As expected, the histogram shows that OpenFlow performs slightly worse than P4 due to the fact that P4 only implements the minimal forwarding behaviour while OpenFlow comes with a set of default features that introduce a not negligible overhead. An important conclusion to draw out of Figure 11 is that *P4DM* does not affect the switch performance, since the P4 switch performs the same either with or without *P4DM*.

## 6. Conclusions

In this work, we have proposed to exploit data plane programmability to measure delays on network links for network management purposes. This is not a new problem, since it was already addressed in the past by the scientific community. The advent of SDN provided new tools that started a new wave of interest on the topic.

In this manuscript, we are along this line. We propose an algorithm based on custom probe packets that is implemented with the P4 language. This allows a very robust implementation, thanks to the local processing of the probes in the switches.

We tested our algorithm along various lines. At first, we compared it with other solutions documented in the related scientific literature. In the manuscript, we show that our algorithm outperforms some of the algorithms already known or, at least has equivalent performance. Then, we have considered the case of the application of the algorithm to multi hop network segments, comparing our proposed solution with the well known ping utility. In the manuscript, we show that *P4DM* proves to be very robust to traffic congestion and packet loss, providing accurate measurement results. Finally, we investigated the load that the algorithm introduces in the switches, showing that it is negligible, therefore not affecting the overall network performance.

Based on the aforementioned results, we believe this work proves the effectiveness of data plane programmability when employed to collect network performance measures, as well as the correctness of the proposed algorithm.

## Figures and Tables

**Figure 1 sensors-22-04411-f001:**
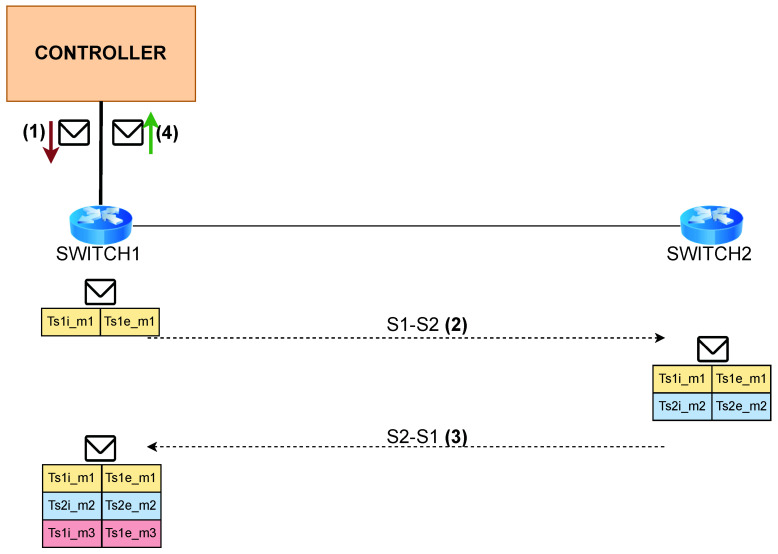
*P4DM* steps.

**Figure 2 sensors-22-04411-f002:**
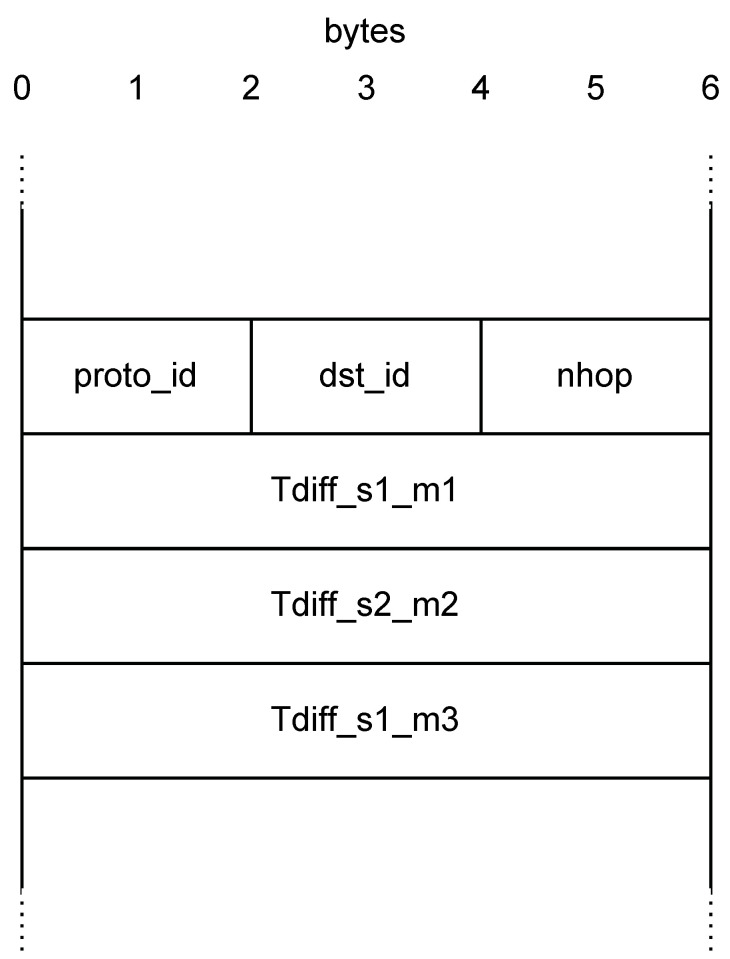
Custom header for *P4DM*.

**Figure 5 sensors-22-04411-f005:**
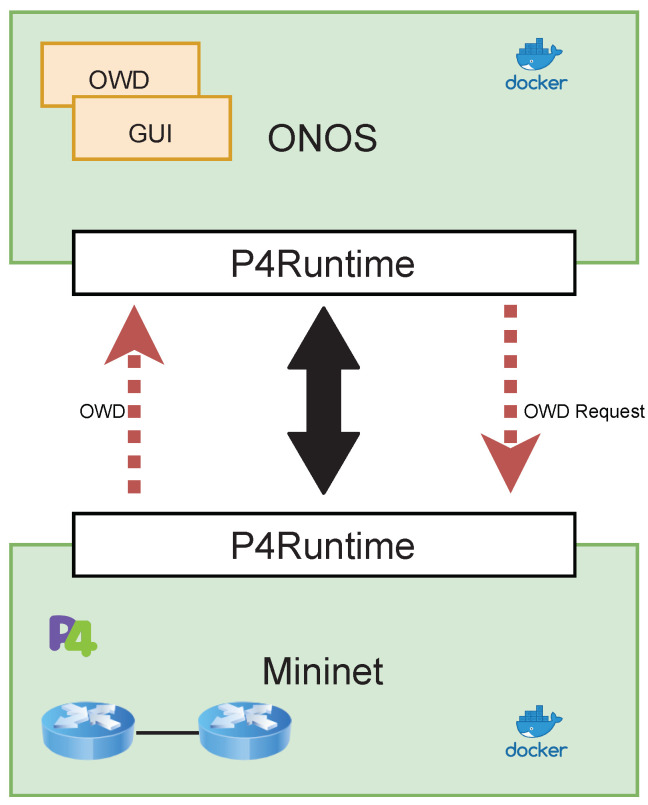
Docker containers infrastructure simulating ONOS-Mininet interaction.

**Figure 7 sensors-22-04411-f007:**
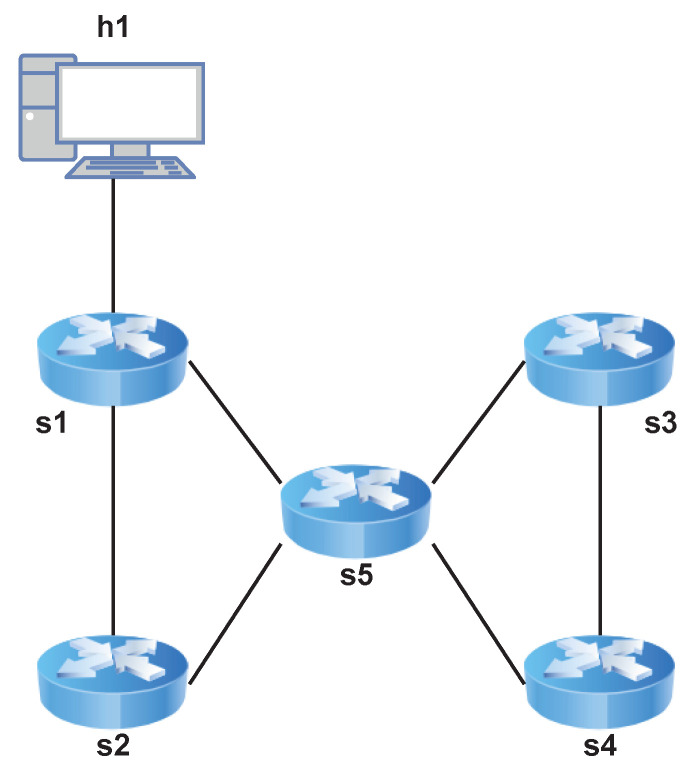
Network for the 4-links path OWD calculation.

**Figure 8 sensors-22-04411-f008:**
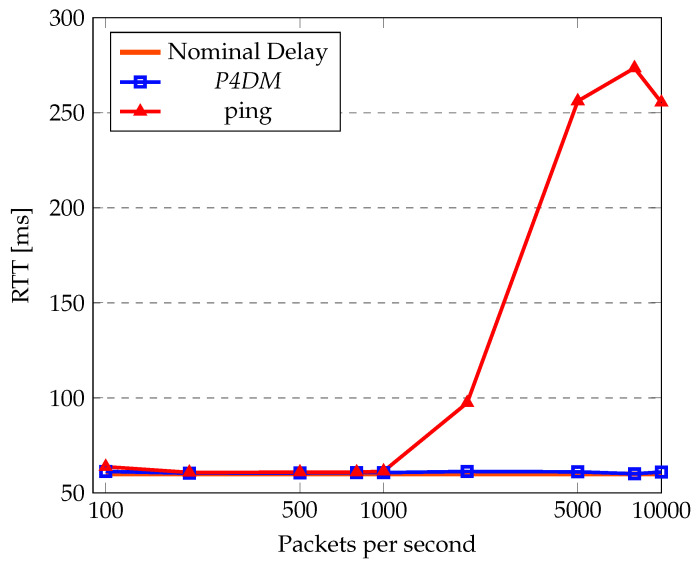
*P4DM* and ping RTT comparison.

**Figure 10 sensors-22-04411-f010:**
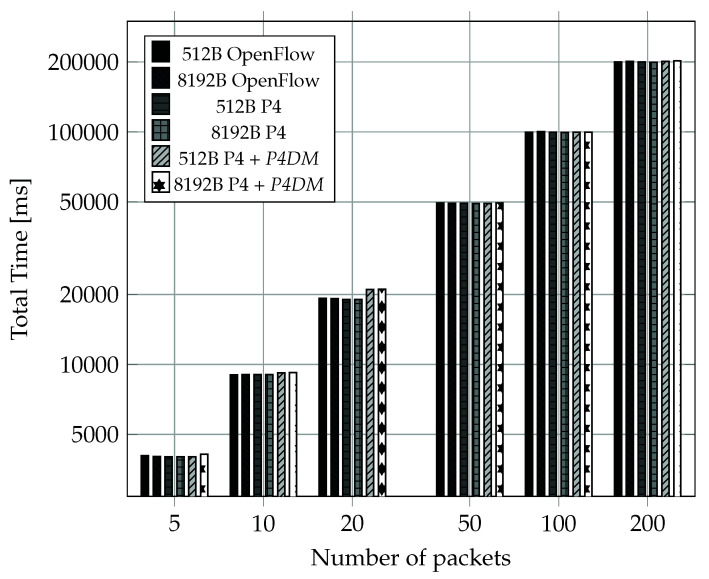
Protocol enabled switches transmission time tests.

**Figure 11 sensors-22-04411-f011:**
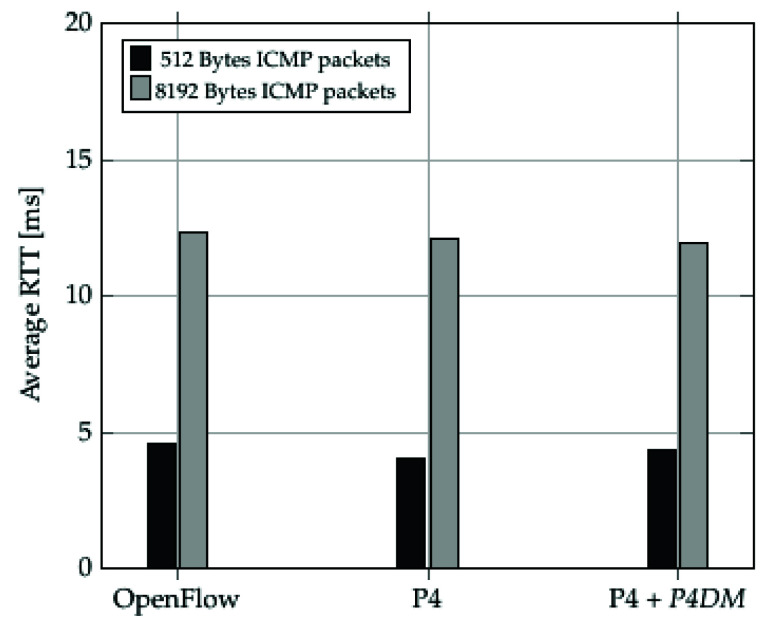
Protocol enabled switches RTT tests.

**Table 1 sensors-22-04411-t001:** Timestamps meaning.

Name	Timestamp Meaning
$T_{s1i\_m1}$	Packet from controller enters ingress queue on switch 1
$T_{s1e\_m1}$	Packet from controller enters egress queue after forwarding decision of sending it towards switch 2
$T_{s2i\_m2}$	Packet from switch 1 enters ingress queue on switch 2
$T_{s2e\_m2}$	Packet enters egress queue on switch 2 after being recognized as a probe to be sent back to switch 1
$T_{s1i\_m3}$	Packet from switch 2 enters ingress queue on switch 1
$T_{s1e\_m3}$	Packet from switch 2 enters egress queue on switch 1 to be sent back to the controller
$T_{diff\_s1\_m1}$	$T_{s1e\_m1}-T_{s1i\_m1}$
$T_{diff\_s2\_m2}$	$T_{s2e\_m2}-T_{s2i\_m2}$
$T_{diff\_s1\_m3}$	$T_{s1e\_m3}-T_{s1i\_m3}$

**Table 3 sensors-22-04411-t003:** Multi-link path performance comparison.

Nominal Delay per Link	1-Link Path Error	4-Links Path Error	5-Links Path Error
10 ms	1.7%	3.5%	3.51%
50 ms	0.31%	1.1%	1.37%
100 ms	0.53%	0.6%	0.52%
200 ms	0.31%	0.26%	0.28%

## Data Availability

The Github repository that contains the code used to gather the results can be found in [33].

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
