# Peer review of "P4DM: Measure the Link Delay with P4"

_sensors, 2022, doi:10.3390/s22124411_

Round 1

Reviewer 1 Report

This paper proposes an approach describing a link delay measurement system to handle lightweight probe packets.

  • I think the presentation of this work must be improved. The abstract is too short and should be extended to describe the problem statement, and the proposed solution, show how the proposed solution is better than other approaches and what are the evaluation metrics used. Also, the methodology and results (Discussion) sections should be improved further in terms of presentation and explain the main steps adopted in a clear manner using graphical abstract at the beginning of the section. Use subsections in sections 4 and 5 to maintain the flow of the work. 
  • I suggest that you rename Section 4  (The proposed implementation) to (The methodology/proposed solution) and rename section 5  (Discussion) to the experimental results.
  • how did you compute the time complexity of the proposed algorithm and other algorithms.
  • In the state of the art section, compare your approach with other works. Also, add some relevant works closed to the proposed solution.
  • You should add a background section that describes the main concepts adopted in the paper.
  • In the results section, summarize the advantages of P4DM and discuss the challenges and considerations.
  • There are some relevant literature review that is not mentioned in the paper such as:https://arxiv.org/pdf/2102.00643.pdf

Reviewer 2 Report

The authors present a P4-based approach to Link Delay Measurements (LMDs) by injecting controller generated packets into the data plane as probes. As the authors state, this approach is not entirely new as related works (many cited) have followed this or a similar approach in the OpenFlow-era of the early 2010s. The application of this approach to P4-style data plane programmability is a logical step. However, it requires the presence of P4-programmable devices in the data plane at ideally every hop. This is unlikely to happen for larger networks and may be limited to certain network elements in enterprise networks or data centers.

In light of that, it is not clear - apart from the obvious advantages of possible P4 data plane performance - what the area of application for this approach would be. At the very least, I suggest to create a more realistic usage scenario to show the reader its applicability.

Furthermore, an evaluation on a single-host using Mininet does not provide reliable numbers. Using software switches competing for the same processing resources amonst each other - and with an SDN controller no less - makes for unreliable measurement results. Minitnet does not offer much in terms of performance guarantees. Therefore, I suggest to evaluate the approach once more using multiple machines and ideally P4-compatible hardware. At this stage, my recommendation is a major revision.

Reviewer 3 Report

This paper examines the issue of Link Delay Measurement (LDM) in Software-Defined Networks that are using P4-enabled switches. The paper explores the use of P4 framework for timestamping probe packets that are subsequently analyzed at the SDN controller for measurement of link delays.

While the paper is well written, technically sound and interesting, there is little novelty in it. The suitability of P4 for network telemetry has already been established in the literature, and the use of timestamping for measuring link delays is also well known. A quick Google Scholar search shows numerous articles that employ this technique (e.g. ref [26] section 4.1). The paper presents interesting experiments in which the performance of P4-based LDM is compared against other methods, however neither the results nor the conclusions provide much novelty.

 The paper starts with a doubtful statement, that measuring link delay still represents an open problem. I'd argue that there are many methods that allow fairly accurate measurement of link delay,several of them noted in the related works section of the paper.

The performance of the proposed scheme is also on par or comparable with other methods, as  figures 7-10 show. As authors noted, the performance of ping suffers when the number of probe packets are increased to 10000/s and higher. I cannot really think of a practical scenario where 10000 probes per seconds need to be sent to the switch. For any practical reasons (100-1000 probes per seconds), all techniques seem to demonstrate relatively similar performance.  So the results here are mainly confirming an existing solution rather than providing a radically different or superior one. 

Table 2 shows the baseline for comparison of the algorithms, and the simulation scenarios seem to vary for different algorithms. One suggestion to improve the quality of comparison is to use the same baseline for all of them. For instance, Controller-in-the-loop and many-data-loop algorithms are tested in an environment with nominal delay of 0-1 ms, which naturally increases the error. The authors should expand their simulation of P4DM to also include the nominal delays  under 10 ms and under 1 ms, as to allow fairer comparison between  these schemes.

Also, on page 4, line 182 "quiet variable" is probably a typo. Did you mean that the delay varies significantly?

Round 2

Reviewer 1 Report

The paper now is much better and I recommend publishing it. 

Reviewer 2 Report

While I find the authors' reasoning on the evaluation method along the lines "everybody does it this way" not a valid argument, I am fully aware of the prices of P4 hardware and therefore will not preclude publication based on this circumstance. Therefore, after the work that has been put in to improve the paper, I have no further objections to publication.